# Investigation of the Reduction in Distributed Acoustic Sensing Signal Due to Perforation Erosion by Using CFD Acoustic Simulation and Lighthill’s Acoustic Power Law

**DOI:** 10.3390/s24185996

**Published:** 2024-09-16

**Authors:** Yasuyuki Hamanaka, Ding Zhu, A. D. Hill

**Affiliations:** 1NSI Technologies, LLC, Tulsa, OK 74136, USA; yhamanaka@nsitech.com; 2Petroleum Engineering Department, Texas A&M University, College Station, TX 77843, USA; danhill@tamu.edu

**Keywords:** distributed acoustic sensing, fiber-optic sensing, hydraulic fracturing, petroleum engineering

## Abstract

Distributed Acoustic Sensing (DAS), widely adopted in hydraulic fracturing monitoring, continuously measures sound from perforation holes due to fluid flow through the perforation holes during fracturing treatment. DAS has the potential to monitor perforation Tulsa, OK 74136erosion, a phenomenon of increasing perforation size due to sand (referred to as proppant) injection during treatment. Because the sound generated by fluid flow at a perforation hole is negatively related to the perforation diameter, by detecting the decay of the DAS signal, the perforation erosion level can be estimated, which is critical information for fracture design. We used a Computation Fluid Dynamics (CFD) acoustic simulator to calculate the acoustic pressure induced by turbulence inside a wellbore and investigated the relationship between the acoustic response from fluid flow through a perforation and the perforation size by running the simulator for various perforation diameters and flow rates. The results show that if the perforation size is constant, the plot between the calculated sound pressure level and the logarithm of flow rate follows a straight line relationship. However, with different perforation sizes, the intercept of the linear relationship changes, reducing the sound pressure level. Lighthill’s power law indicates that the change in intercept corresponds to the logarithm of the ratio of the increased diameter to the original diameter. The reduction in sound pressure level observed in the CFD simulation correlates with the reduction in the DAS signal in field data. The findings of this study help to evaluate perforation diameter growth using DAS and interpret fluid distribution in fracture stimulation.

## 1. Introduction

This paper is a continuous study of the conference paper [1], providing a more in-depth discussion of the simulation results and additional modeling study.

Distributed acoustic sensing (DAS) is a widely used fiber optic technology that records acoustic vibration along the length of a fiber-optic cable. In the oil and gas industry, DAS is utilized to monitor fracture growth during hydraulic fracture treatments. When DAS is deployed in treating wellbore, it continuously measures the sound emitted from perforation holes (a group of perforation holes is referred to as a perforation cluster) due to fluid flow from the wells through the holes to the reservoir. The acoustic responses provide valuable insight into the dynamically changing fluid distributions throughout the treatment [2,3,4]. Based on signal amplitude, DAS has been used to estimate flow rates at individual clusters [5,6,7,8]. This information helps to understand the fluid distribution and is used to estimate fracture geometry based on the volume of fluids into each cluster.

However, discrepancies arise when comparing DAS results with other diagnostic tools, such as Distributed Temperature Sensing (DTS) [7,9] and surface treating pressure [10]. One possible reason is that the current DAS interpretation methodologies do not account for signal decay due to perforation erosion. Perforation erosion occurs when slurry (sand mixed with fracture fluid to support the fracture opening after hydraulic fracturing) impacts or scraps against the perforation entrance. Because the acoustic energy generated by flow is correlated to the size of the perforation hole. As the perforations erode and the holes get bigger, the frictional pressure drop through perforation friction reduces and the acoustic energy of the perforation flow decreases, resulting in weaker sound [10]. Consequently, DAS signal interpretation without correcting the perforation sound deviates from the true fluid distribution. 

Figure 1 is a typical DAS waterfall plot used in monitoring fracturing. In the 3D contour plot, the vertical axis is the distance along a wellbore, the horizontal axis is treating time, and the contour color represents the intensity of the DAS signal, which is related to the volume of fluid at a certain location and specific time. As shown in Figure 1, the DAS signal decays halfway through the treatment and disappears by the end of the treatment, despite the injection rate being constant throughout the treatment. Cramer and Zhang [10] mention that the decay of the signal is due to perforation erosion. Somanchi et al. [9] also observed the fading in DAS signals due to perforation erosion. They mention that the acoustic signal shifts into lower frequency bands as perforations erode, leading it to look like fading.

Perforation erosion poses a significant challenge for hydraulic fracturing. Optimized fracture designs aim at generating a uniform distribution of fractures along the entire horizontal well. During hydraulic fracturing, initially eroded perforations receive more slurry due to the reduced frictional pressure, causing further erosion. In contrast, perforations that are not eroded have high frictional pressure, preventing fluid entry. Consequently, by the end of the treatment, only a portion of perforations that have eroded are effectively stimulated, leading to lower fracture efficiency. The decay of the DAS signal has the potential to serve as an indicator of perforation erosion and helps engineers assess cluster efficiency. 

Vortices caused by multiphase flow are studied across various industries, including aerospace, nuclear power, and manufacturing, using Computational Fluid Dynamics (CFD) modeling. In [11], the acoustics induced by a multiphase sink vortex propagating through multiple acoustic media was modeled by coupling CFD with fluid–solid acoustic theory. Additionally, ref. [12] provides a comprehensive review of the Soft Abrasive Flow (SAF) finishing method, emphasizing modeling efforts that investigate the interaction between abrasive particle flow and the workpiece using CFD. 

The objective of this study is to develop new relationships of DAS interpretation for fracture diagnosis with consideration of perforation erosion. It investigates the differences in the acoustic response of flow-induced sound due to varying perforation sizes using a CFD acoustic model. The CFD results are compared with previous experimental studies and theoretical solutions, leading to the development of a theoretical model that illustrates the effects of perforation erosion on DAS signals.

## 2. Literature Reviews

### 2.1. Modeling of Acoustics Induced by Near-Wellbore Flow

Mckinley et al. [13] investigated the sound induced by fluid leakage from a formation to a wellbore through the channels in cement. Their laboratory-scale experiment and acoustic theory are also applicable to sound induced by perforation flow. They constructed a leak simulator consisting of a casing and throttle, with water or air injected from the throttle into the casing. The sound induced at the throttle was measured by an acoustic sonde located in the casing. The experiment confirmed that acoustic amplitude is linearly correlated with the energy dissipation rate at the throttle as follows:(1)$$A_{n}=F\left(∆P_{t}q\right)=F\left(C_{D}q^{3}\right),$$
where $A_{n}$ is the sound amplitude, $∆P_{t}$ is the friction pressure drop at the throttle, q is the flow rate at the throttle, and $C_{D}$ is the drag coefficient. Since the friction pressure drop at the throttle is a function of the square of the flow rate, the amplitude correlates with the cube of the flow rate.

In 2016, Stockley [14] conducted large-scale flow loop testing to record acoustic responses generated by fluid flowing through perforations in a long horizontal steel pipe. The acoustic signature was recorded by a fiber-optic acoustic sensor cable attached along the entire length of the flow loop. The acoustic signature was used to develop a semiempirical acoustic-based flow model. Shen et al. [5] applied the model to field data to evaluate quantitative flow rate at individual clusters in real time.

Chen et al. [15] constructed a laboratory-scale wellbore system with fracture cells filled with proppant. The wellbore and the fracture cells were connected with small pipes simulating perforations. They injected fluid from the tip of the fracture cell to the wellbore and measured the sound generated at the perforation tunnels, evaluating the effects of proppant concentration and size on acoustic signals. Pakhotina et al. [16] built the same wellbore geometry model using commercial CFD software and successfully reproduced the experimental results. Based on both experimental and numerical trials, they obtained the following correlation:(2)$$\log\left(q^{3}\right)=A\times L_{sp}+B,$$
where q is the flow rate at a perforation, $L_{sp}$ is the measured sound pressure level, and A and B are the coefficients of the linear correlation. Equation (2) is consistent with Equation (1), as both describe acoustic behavior as a function of the cube of the flow rate. Although, this correlation was originally developed assuming the production phase, Pakhotina [6] demonstrated that it also applies to flow through perforations during the injection phase. Based on field DAS data from a fracturing stage with a single cluster and no proppant injection, Sakaida et al. [7] observed that this linear correlation remains valid when the sound pressure level is replaced by the frequency band energy, which is the energy response of the DAS signal. 

### 2.2. Lighthill’s Acoustic Analogy

Lighthill considered a situation where sound is generated in a turbulent flow region and then propagates to an observer through a surrounding medium with uniform density and sound speed [17]. He analytically described the sound generation due to turbulence by transforming the Navier–Stokes and continuity equations:(3)$$\frac{1}{c_{0}^{2}}\frac{\partial^{2}p^{\prime }}{\partial t^{2}}-\nabla^{2}p^{\prime }=\frac{\partial^{2}T_{ij}}{\partial x_{i}\partial x_{j}},$$
where $c_{0}$ is the uniform sound speed, $p^{\prime }$ is the sound pressure, t is the time, and $T_{ij}$ is Lighthill’s stress tensor. The left side of the equation represents the propagation of the sound pressure $p^{\prime }$ in the space with uniform density and sound speed around the turbulence. The right side represents the sound source generated within the turbulence, defined as follows:(4)$$T_{ij}=\rho u_{i}u_{j}+\left\{(p-p_{0})-c_{0}^{2}\left(\rho -\rho_{0}\right)\right\}\delta_{ij}-\sigma_{ij}.$$
where ρ is the density, $\rho_{0}$ is the mean density, p is the pressure, $p_{0}$ is the mean pressure, $\delta_{ij}$ is the Kronecker delta, and $\sigma_{ij}$ is the viscous stress. The first term, $\rho u_{i}u_{j}$, represents the sound source due to the convective transport of fluid flow and is significant only where the flow is turbulent. The second term $\left\{(p-p_{0})-c_{0}^{2}\left(\rho -\rho_{0}\right)\right\}\delta_{ij}$ represents the sound source due to the excess momentum transfer between the turbulence and the ideal fluid with constant density assumed in the acoustic field outside the turbulence. The third term $\sigma_{ij}$ represents the acoustic attenuation due to the viscosity.

When the flow is incompressible, meaning the characteristic Mach number, $M\sim v/c_{0}$, is small ($M^{2}\ll 1$), the term due to density variation, $\left(p-p_{0}\right)-c_{0}^{2}\left(\rho -\rho_{0}\right)$, in Lighthill’s stress tensor (Equation (4)) is negligible [16]. Therefore, Lighthill’s stress tensor contributing to the sound source at low Mach numbers is as follows:(5)$$T_{ij}\approx \rho_{0}u_{i}u_{j}+\sigma_{ij},\;\;when\;M^{2}\;\ll 1.$$

Ignoring the viscous term of the Lighthill equation, solving Lighthill’s equation (Equation (3)) in the far-field conditions, and performing dimensionless analysis, Lighthill’s acoustic power law is derived [16].
(6)$$p^{\prime }\sim \frac{D}{\left|x\right|}\frac{\rho_{0}U^{4}}{c_{0}^{2}}$$
where D is the characteristic length, U is the characteristic velocity, and $\left|x\right|$ is the distance between the turbulence and the observer. Lighthill applied this equation for the investigation of the acoustics emitted from a jet engine, defining the characteristic length as the jet diameter and the characteristic velocity as the jet velocity. The acoustic power law is well confirmed with the experiments and field data [18]. 

Since the equation assumes an incompressible condition and perforation flow is often modeled as jet flow [19], Lighthill’s acoustic power law is likely applicable to the DAS signal generated by the perforation flow. 

## 3. CFD Acoustic Simulation Procedure

A numerical simulation integrating CFD and aerodynamics is used to investigate the relationship between perforation size and the acoustic response induced by the perforation flow. This simulation calculates the acoustic pressure generated by turbulence at arbitrary located acoustic receivers outside the fluid flow domain. The commercial CFD software ANSYS Fluent 2021 R2 is used for the simulation. The simulation results lay the fundamentals for considering the decay of the DAS signal due to perforation erosion. 

To initiate the calculation, a physical geometry consisting of fluid domains inside a perforation and a wellbore is built, as shown in Figure 2a. Due to the symmetry of the problem, half the geometry is constructed, as shown in Figure 2b, to improve computational efficiency. Water is injected from one end of the wellbore and drained out from the perforation outlet, while the other side of the wellbore is closed. The geometry is discretized into a mesh, as shown in Figure 3. A fine mesh is used only at the perforation because the eddies generated there are smaller than those in the wellbore domain. Appropriate mesh size and timestep size are investigated in the next chapter using the small-scale wellbore model. 

Once the mesh is generated, the simulation proceeds to the CFD calculation part, which adopts a coupled method to estimate the acoustic pressure. The calculation for fluid field utilizes Large Eddy Simulation (LES) to calculate the fluctuating component of turbulence. Ffowcs Williams–Hawkings model (FW-H model) simultaneously calculates the acoustic source information based on the turbulent profile [20]. The acoustic pressure at each receiver is calculated every time step based on the acoustic source information by ANSYS Fluent. 

### 3.1. Turbulent Flow Calculation

LES numerically solves only the large eddies on grids, while the small eddies are modeled to maintain a balance between the accuracy and the computational cost. Therefore, LES can capture unsteady and complex turbulent perturbations at a reasonable computational cost.

LES filters out the eddies smaller than the mesh size. The filtered variable is defined as follows:(7)$$\tilde{\varphi }\left(x\right)=\frac{1}{V}\int_{D}\varphi \left(x^{\prime }\right)dx,\;if\;x^{\prime }\in \nu ,$$
where $\tilde{\varphi }$ is the filtered variables of φ, V is the volume of a computational cell, and x is the coordinate. Both conservation of mass and conservation of momentum are represented using the filtered variables:(8)$$\frac{d\rho }{dt}+\frac{\partial }{\partial x_{i}}\left(\rho \tilde{u_{i}}\right)=0,$$
(9)$$\frac{d}{dt}\left(\rho \tilde{u_{i}}\right)+\frac{\partial }{\partial x_{j}}\left(\rho \tilde{u_{i}}\tilde{u_{j}}\right)=-\frac{\partial \tilde{p}}{\partial x_{j}}+\frac{\partial }{\partial x_{j}}(\sigma_{ij}-\tau_{ij}),$$
where t is the time, ρ is the fluid density, u is the fluid velocity, p is the static pressure, $\sigma_{ij}$ is the viscous stress tensor, and $\tau_{ij}$ is the subgrid-scale stress tensor. The viscous stress tensor is also defined by using the filtered variables:(10)$$\sigma_{ij}=\mu \left[\left(\frac{\partial \tilde{u_{i}}}{\partial x_{j}}+\frac{\partial \tilde{u_{j}}}{\partial x_{i}}\right)-\frac{2}{3}\frac{\partial \tilde{u_{i}}}{\partial x_{i}}\delta_{ij}\right].$$
where μ is the viscosity and $\delta_{ij}$ is the Kronecker delta. The subgrid-scale stress tensor $\tau_{ij}$ represents the resistive force that the filtered-out eddies exert. Since it cannot be solved by the mesh, there are several models to estimate the value. The Smagorinsky–Lilly model [21] is utilized in this research. 

### 3.2. Ffowcs Williams–Hawkings Model

Ffowcs Williams and Hawkings extended Lighthill’s acoustic analogy to account for the sound generated by the interaction between turbulence and solid surfaces. The FW-H equation is as follows:(11)$$\begin{array}{c} \frac{1}{c_{0}^{2}}\frac{\partial^{2}p^{\prime }}{\partial t^{2}}-\nabla^{2}p^{\prime }=\frac{\partial }{\partial t}\left[\left\{\rho_{0}v_{j}+\rho \left(u_{j}-v_{j}\right)\right\}\frac{\partial H}{\partial x_{j}}\right]\\ \;\;\;\;\;\;\;\;\;\;\;\;\;\;\;\;\;\;\;\;\;\;\;\;\;\;\;\;\;\;\;\;\;\;\;\;\;\;\;\;\;\;\;\;\;\;\;\;\;\;\;\;\;\;\;\;\;-\frac{\partial }{\partial x_{i}}\left[\left\{\rho u_{i}\left(u_{j}-v_{j}\right)+\left(p-p_{0}\right)\delta_{ij}-\sigma_{ij}\right\}\frac{\partial H}{\partial x_{j}}\right]\\ \;\;\;\;\;\;+\frac{\partial^{2}}{\partial x_{i}\partial x_{j}}\left\{T_{ij}H\right\}. \end{array}$$
where v is the velocity of control surface, $T_{ij}$ is Lighthill’s stress tensor, and H is the Heaviside function. 

The first and second terms on the left side of Equation (5) represent the sound source due to the interaction between turbulence and surface; they are nonzero only on the surface. The third term represents the sound generated by the turbulence. 

Ansys Fluent employs Ffowcs Williams and Hawkings model, enabling us to calculate sound generated at the perforation wall, which Lighthill’s equation cannot account for, as well as sound induced by turbulence. 

### 3.3. Acoustic Signal Processing

After the acoustic pressure is calculated at the receiver, it is converted to the frequency spectrum by the Fast Fourier Transform (FFT). Then, the auto-spectral density is calculated as follows:(12)$$S\left[m\right]=\frac{2X*\left[m\right]X\left[m\right]}{\alpha^{2}},\left(m=0,1,2,\cdots ,\frac{N}{2}\right),$$
where X is the signal after the acoustic pressure is converted to the frequency domain, ∗ represents the complex conjugate, N is the number of samples, $\alpha =\sum_{n=0}^{N}w[n]$, and w is the window. The auto-spectral density in the form of the sound pressure level $L_{sp}$ is used to evaluate the frequency spectrum:(13)$$L_{sp}[m]=10\log\left(\frac{S[m]}{p_{ref}^{2}}\right),m=0,1,2,\cdots ,f_{s}/2,$$
where $p_{ref}$ is the reference sound pressure and $p_{ref}^{\prime }=2\times 10^{-5}\;Pa=2.90\times 10^{-9}\;psi$. The overall sound pressure level, which is the summation of the sound pressure level in a particular frequency band $L_{sp_{overall}}$ is calculated by the following equation:(14)$$L_{sp_{overall}}=10\log\left(\sum_{m=f_{min}}^{f_{max}}10^{\frac{L_{sp}\left[m\right]}{10}}\right),$$
where $f_{min}$ is the minimum frequency in the band and $f_{max}$ is the maximum frequency in the band.

## 4. Determination of Time Step Size by Small-Scale Simulation

Since time step size and mesh size are crucial for the accuracy of the acoustic calculation, we ran the small-scale CFD acoustic simulation multiple times and decided the sampling frequency, which is the inverse of the time step size. The physical geometry consists of fluid domains inside a wellbore and a perforation, as shown in Figure 4a. Because the domain is symmetric, half of the geometry is built, and a symmetry boundary condition is applied. The wellbore is 1 inch (2.54 cm) long and 2 inches (5.08 cm) inner diameter. The perforation is 2 inches (5.08 cm) long, and the diameter is 0.3 inches (0.76 cm). The perforation is located at the center of the wellbore. Then, 1.5 bbl/min (0.0040 m^3^/s) of water is injected from one side of the wellbore, resulting in perforation friction of 1000 psi ($6.9\times 10^{6}$ Pa). Since the other side of the wellbore is closed, all fluid is forced to flow into the perforation. Then, 6000 psi ($4.1\times 10^{7}$ Pa) of backpressure is applied on the perforation outlet as bottomhole fracture pressure. To make the calculation simple, the wall friction is ignored. The sound speed is set to 1115 ft/s (343 m/s), which is the same speed in air. Because the FW-H model assumes acoustically far-field, the receiver is placed one wavelength far from the fluid domain. We placed the receiver at 11.15 ft (3.39 m) directly below the perforation entrance, and therefore the lowest frequency is 100 Hz (=1115 ft⁄s/11.15 ft). The receiver is placed at 11.15 ft directly below the perforation entrance, and therefore the lowest frequency is 100 Hz (=1115 ft⁄s/11.15 ft). 

Figure 4b shows the meshed small-scale wellbore model. To reduce the computational cost, the perforation domain is discretized into finer elements and the wellbore domain is discretized into coarser elements.

To stabilize the transient flow calculation, the mesh size is adjusted according to the sampling frequency. This adjustment is necessary because numerical simulations discretize partial differential equations (PDEs) in both time and space. To ensure accuracy, fluid particles must not skip grid points within a single time step, adhering to the Courant–Friedrichs–Lewy (CFL) condition. We tried six cases with different mesh sizes. The sampling frequencies are shown in Table 1. The corresponding time step sizes are also shown in the table. We calculated the acoustic pressure for 0.1 s. Thus, the frequency resolution is 10 Hz (=1/0.1 s).

Because the sampling frequency of DAS is typically 16,000 Hz and the maximum measurable frequency (a half of the sampling frequency) is 8000 Hz, we focus on the acoustic signals below 8000 Hz. The calculated acoustic signal in the DAS frequency range is shown in Figure 5. The spectra with the sampling frequency less than 1428 kHz are significantly underestimated than those with higher sampling frequencies because of aliasing error. The spectra have similar shapes with sampling frequencies higher than 1666 kHz. 

To decide the appropriate sampling frequency, the average sound pressure level between 100 Hz and 8000 Hz at different sampling frequencies is obtained and plotted in Figure 6. As the sampling frequency increases, the change in the average sound pressure becomes smaller, indicating convergence. In terms of the computational cost and accuracy, we decided to use a sampling frequency of 1666 kHz or higher, which corresponds to a time step size of less than 6.0 × 10^−7^ s. This also implies that DAS, with a sampling frequency significantly lower than 1.666 kHz, may not adequately capture the acoustic characteristics in a higher frequency range.

## 5. Acoustic Simulation with Large-Scale Simulation

After the sampling frequency and the time step size are determined, the large-scale simulations are performed to evaluate the effect of perforation size on the amplitude of flow-induced sound around a perforation. The large-scale simulation, similar to the small-scale simulation, consists of a perforation region and a wellbore region, as shown in Figure 1 and Figure 2. The inner diameter of the wellbore is 4.77 inches (12.1 cm), and its length is 1 ft (0.305 m). The perforation is placed at the center of the wellbore (0.5 ft (0.253 m) from the inlet). Half of the geometry is built, and a symmetric boundary condition is applied. To simulate the bottom hole fracture pressure, 6000 psi ($4.1\times 10^{7}$ Pa) is applied on the perforation outlet. Because we want to compare perforation friction pressure and flow-induced sound, wall friction is neglected. 

The sound speed is set to 1115 ft/s (334 m/s), and the receiver is placed at 11.15 ft (3.39 m) directly below the perforation entrance. If we use the speed of sound in water, which is 4900 ft/s (1500 m/s), the receiver will need to be located 49 ft (15 m) far from the wellbore to calculate 100 Hz. Therefore, in this study, the sound speed in air is chosen to minimize the distance between the receiver and sound source. Thus, the lowest frequency is 100 Hz (=1115 (ft/s)/11.15 (ft)). The calculation time is limited to 0.1 s due to the computational cost, resulting in a frequency resolution of 10 Hz (=1/0.1 (s)).

We generate computational geometries with perforation diameters of 0.25 inches (0.64 cm), 0.3 inches (0.76 cm), and 0.35 inches (0.89 cm) to investigate the effect of perforation size on sound induced by perforation flow. We simulate the flow conditions at three different injection rates, respectively, to set the target perforation friction to 1000 psi ($6.9\times 10^{6}$ Pa), 1500 psi ($1.0\times 10^{7}$ Pa), and 2000 psi. The injection rates are determined using the following orifice equation [18]:(15)$$∆p_{perf}=\frac{0.2369\rho q^{2}}{N^{2}C_{d}^{2}d^{4}},$$
where $∆p_{perf}$ is the perforation friction pressure, N is the number of perforations, $C_{d}$ is the discharge coefficient, and d is the perforation diameter. The calculation conditions are summarized in Table 2. The sampling frequencies for 0.25 inches (0.64 cm) and 0.3 inches (0.76 cm) perforation are 2500 kHz, while the sampling frequencies for 0.35 inches (0.89 cm) perforation are 2000 kHz. This is because the larger perforation diameter generates less turbulence at the perforation entrance, making the calculation easier to converge with a simpler computational condition. 

The static pressure and the dynamic pressure calculated by LES are presented in Figure 7 and Figure 8, respectively. These results correspond to the calculation with a perforation diameter of 0.3 inches (0.76 cm) and the injection rate of 1.54 bbl/min (0.0041 m^3^/s), resulting in the target perforation friction of 1000 psi ($6.9\times 10^{6}$ Pa). Since this calculation neglects the pipe friction, the static pressure change occurs only at the perforation entrance.

The calculated acoustic spectra are shown in Figure 9a–c. The sound pressure level from 100 Hz to 8000 Hz is selected to represent the DAS frequency band. The sound pressure level drops markedly at 3400 Hz and 6400 Hz in all the calculations, but this is due to the resonance caused by the acoustic boundary conditions created by the geometry wall. 

Figure 9a–c do not exhibit clear peaks within the DAS frequency band for any perforation sizes, and the spectra have symmetrical shapes. However, the empirical results published in [13], which examined sound induced at a throttle, demonstrated peak frequencies shifting from 2000 Hz to 1000 Hz as the throttle pressure decreased, as shown in Figure 9d. Furthermore, ref. [9] observed frequency shifts in field DAS data due to perforation erosion. One reason for the discrepancies with the previous studies could be the small perforation sizes used in the simulation. This is explained by Lighthill’s dimensionless analysis, which relates the characteristic frequency, F~U/D where U is the characteristic velocity and D is the characteristic length. In our situation, the characteristic velocity is the velocity at the perforation entrance, and the characteristic length is the perforation diameter. Therefore, the minimum characteristic frequency is 10,114 Hz when the perforation pressure is 1000 psi ($6.9\times 10^{6}$ Pa) and the perforation diameter is 0.35 inches (0.89 cm). It is possible that certain peaks were not observed because the characteristic frequencies lie outside the DAS frequency range. However, if the perforation diameter increases or the flow velocity decreases, the characteristic frequency will decrease, and a peak may be observed within the DAS frequency range.

Another potential reason for the discrepancies is that the simulation assumes the acoustic waves propagate through a uniform fluid surrounding the turbulence. In contrast, in the field or experiment, the induced sound propagates through the media with various densities and sound speeds, such as pipes, cement, and formation, which could cause frequency change. 

To compare the relationship between flow rate and sound induced by the perforation flow, the overall sound pressure levels from 100 Hz to 8000 Hz are calculated using Equation (14). Figure 10 shows the relationship between $\log\left(q\right)$ and the overall sound pressure level for various perforation sizes. If the perforation size is constant, the overall sound pressure level is linearly correlated with $\log\left(q\right)$. This result is consistent with Equation (2) when the power of 3 is decomposed and included in the slope and the intercept. However, when the perforation size changes, the slope and intercept of the linear relationship are found to be different.

To verify McKinley’s assumption that flow-induced noise results from energy dissipated by fluid turbulence (Equation (1)), Figure 10b compares the logarithm of the perforation flow energy defined as $\log\left(∆p_{perf}q\right)$ and the overall sound pressure level. A consistent linear correlation between $\log(∆p_{perf}q)$ and the overall sound pressure level is observed. The plots show that even though the linear relationship is still valid, the effect of perforation size is reduced because the perforation pressure drop is a function of the perforation diameter. The three perforation size plots are following the same slope, with 0.25 inches plots slightly below the plots for larger-sized perforations. 

We define the characteristic length D as the perforation diameter d and the characteristic velocity U as the fluid velocity at perforation $v_{perf}$. Because the density, sound speed, and distance between the receiver and the perforation are constant, the acoustic pressure $p^{\prime }$ and sound pressure level $L_{sp}$ are represented with the following equation based on Lighthill’s power law (Equation (6)).
(16)$$p^{\prime }\sim \frac{d}{\left|x\right|}\frac{\rho_{0}v_{perf}^{4}}{c_{0}^{2}}\sim dv_{perf}^{4},$$
(17)$$L_{sp}=20\log\left(\frac{p\prime }{p_{ref}}\right)\sim \log\left(dv_{perf}^{4}\right).$$

Figure 10c uses $\log\left(v^{4}d\right)$ as a correlating parameter for the overall sound pressure level. A consistent linear correlation between them is observed. Moreover, 0.25 inches (0.64 cm) plots are on the line, and it is better than McKinley’s assumption. 

In summary, this finding can be used in the field to evaluate perforation erosion. Before a perforation hole is significantly eroded, with different flow rates passing the perforation, the DAS data plot as shown in Figure 10a should move along a straight line with consistent slope and intercept. Once the perforation size is enlarged, the intercept changes, and the data points should align with a new straight line. The slope of the new line may not change much, but the intercept increases. The change in intercept serves as an indicator of perforation erosion. 

## 6. Perforation Flow-Induced Acoustic Model

Based on the numerical simulation results, a theoretical perforation flow-induced acoustic model is developed. Figure 10c indicates that Lighthill’s acoustic power law, Equation (16), is valid for the sound induced by the perforation flow. Thus,
(18)$$\log\left(v_{perf}^{4}d\right)=A_{o}\times 20\log\left(\frac{p^{\prime }}{p_{ref}}\right)+B_{o},$$
(19)$$\log\left(\frac{{2.746\times 10^{3}q}^{4}}{C_{d}^{4}d^{7}}\right)=A_{o}L_{sp}+B_{o}$$
where the coefficient $2.746\times 10^{3}$ is the unit convergence, q is the flow rate at the perforation in bbl/min, d is the perforation diameter in inches, and $A_{o}$ and $B_{o}$ are the slope and intercept of Figure 10c.

Before the perforation is eroded, the perforation has the initial diameter $d_{ini}$, which is 0.25 inches (0.64 cm) in our simulation. The initial diameter $d_{ini}$ can be substituted into Equation (19), then
(20)$$\log\left(\frac{{2.746\times 10^{3}q}^{4}}{C_{d}^{4}d_{ini}^{7}}\right)=A_{o}L_{sp}+B_{o}$$

Decomposing and moving the left side to the right side, it is transformed into the correlation in terms of $\log\left(q\right)$, which is shown in Figure 10a as follows:(21)$$\log\left(q\right)=\frac{A_{0}}{4}L_{sp}+\frac{1}{4}\left\{B_{0}-\log\left(\frac{2.746\times 10^{3}}{C_{d}^{4}}\right)+7\log\left(d_{ini}\right)\right\}$$
(22)$$\log\left(q\right)=AL_{sp}+B_{ini}$$
where $A=A_{o}/4$, which is the slope, and $B_{ini}=\left\{B_{o}-\log\left(2.746\times 10^{3}/C_{d}^{4}\right)+7\log\left(d_{ini}\right)\right\}/4$, which is the intercept before perforation erosion (blue line of Figure 10a).

After the perforation is eroded, the perforation diameter becomes 0.3 inches (0.76 cm) or 0.35 inches (0.89 cm) in the simulation. Even when the perforation is eroded, the correlation between $\log\left(v_{perf}^{4}d\right)$ and $L_{sp}$ has the same slope and intercept, as shown in Figure 10c. Therefore, Equation (19) also works for the eroded diameter $d_{e}$ as follows: (23)$$\log\left(\frac{{2.746\times 10^{3}q}^{4}}{C_{d}^{4}d_{e}^{7}}\right)=A_{o}L_{sp}+B_{o}$$

Again, decomposing and moving the left side to the right side, the correlation in terms of $\log\left(q\right)$ has the same slope but different intercepts as follows:(24)$$\log\left(q\right)=\frac{A_{0}}{4}L_{sp}+\frac{1}{4}\left\{B_{0}-\log\left(\frac{2.746\times 10^{3}}{C_{d}^{4}}\right)+7\log\left(d_{3}\right)\right\}.$$

To arrange the equation, $7/4\log\left(d_{ini}\right)$ is added and subtracted on the right side as follows:(25)$$\log\left(q\right)=\frac{A_{0}}{4}L_{sp}+\frac{1}{4}\left\{B_{0}-\log\left(\frac{2.746\times 10^{3}}{C_{d}^{4}}\right)+7\log\left(d_{3}\right)\right\}.$$
(26)$$\log\left(q\right)=AL_{sp}+B_{ini}+\frac{7}{4}\log\left(\frac{d_{e}}{d_{ini}}\right).$$

This equation indicates that $\log\left(q\right)$ and the sound pressure level linearly correlate even if the diameter changes. After the perforation is eroded, the intercept changes logarithmically with the ratio of the eroded diameter to the initial diameter. The slope of the correlation remains constant despite perforation erosion. 

The empirical correlation, Equation (2), indicates that $\log\left(q^{3}\right)$ is the correlation parameter for the sound pressure level. However, based on the above derivation, $\log\left(q^{4}\right)$ should be linearly correlated with the sound pressure level. Despite this discrepancy, the conclusions of both empirical studies and this theoretical study are consistent because both models use a logarithmic scale, and the power term is incorporated into slope A. 

Because slope A is constant even if the diameter changes, as indicated by the derivation, the linear lines between $\log\left(q\right)$ and the overall sound pressure level shown in Figure 10a are calculated again with the constant slope and illustrated in Figure 11. The solid line represents the linear correlation based on the numerical simulation. The dotted line is the correlation that compensates for the effect of perforation erosion on the 0.25 inches (0.64 cm) result (solid blue line) using Equation (26). Although the dotted lines are slightly below the solid lines, the maximum error is 8.1%, which is acceptable. A possible cause of the slight error is that the correlation does not consider the sound generated in the wellbore, but numerical simulations do.

Figure 12 shows the sensitivity of the derived acoustic correlation (Equation (26)). It illustrates the reduction in the sound amplitude due to perforation erosion, leading to the decay of the DAS signal. For example, if the injection rate is 1 bbl/min (blue line), the sound pressure level is 120 dB for a perforation diameter of 0.3 inches (0.76 cm). However, after the perforation is eroded and the diameter increases to 0.6 inches (1.5 cm), the sound pressure level is reduced to 110 dB. 

## 7. Conclusions

This study investigates the relationship between perforation size and DAS signal amplitude as affected by perforation erosion by modeling the sound induced by perforation flow using the CFD simulation. The simulation results are used to explore the correlation between sound pressure level and three key parameters: $\log\left(q\right)$, $\log\left(∆p_{perf}q\right)$, and $\log\left(v_{perf}^{4}d\right)$. Finally, a theoretical equation was derived from Lighthill’s power law to express the relationship between perforation diameter and sound pressure level, which explains the decay of the DAS signal due to perforation erosion. Future studies may yield further insights by considering the frequency changes caused by perforation shape, slurry additives such as proppant and friction reducer, and the cement and formation around the wellbore. The following list summarizes the conclusions from this study.The frequency spectrum remains symmetrical within the range of 100 Hz to 8000 Hz, with no specific frequency peaks observed, regardless of perforation diameter, as long as the perforation size is less than 0.35 inches. However, frequency peaks may be observed if the perforation diameter is further increased.Although $\log\left(q^{3}\right)$ was used as a correlation parameter in the experimental equation, $\log\left(q^{4}\right)$ should correlate with sound pressure level based on the simulation results and Lighthill’s acoustic power law. The power of $\log\left(q^{4}\right)$ is decomposed, leading to an acoustic correlation: $\log\left(q\right)=A\times L_{sp}+B$.The slope A of the acoustic correlation should be constant even with changes in perforation diameter. For a given perforation size, the acoustic correlation has a unique intercept B. As the perforation size increases, the intercept B increases logarithmically with the ratio of the increased diameter to the original diameter, resulting in a reduction in sound amplitude.

## Figures and Tables

**Figure 1 sensors-24-05996-f001:**
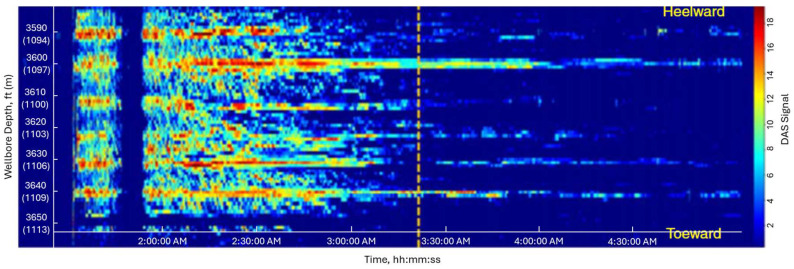
Decay of the DAS signal due to the perforation erosion (after [9]). Color indicates amplitude of DAS signal; x axis indicates the time; and y axis indicates the wellbore depth. The strong signals are observed at the depth of clusters. However, the signal amplitude reduces in the middle of the treatment due to perforation erosion.

**Figure 2 sensors-24-05996-f002:**
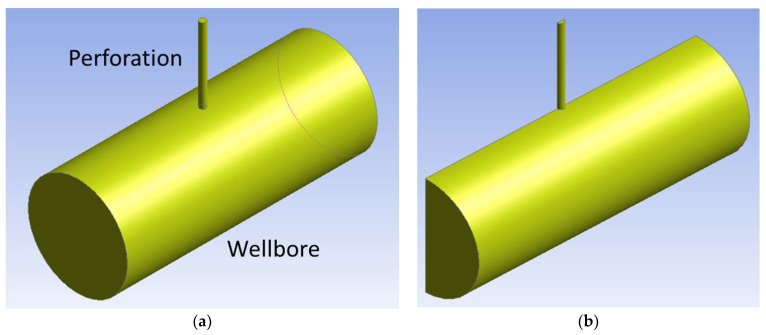
Physical geometry of the CFD simulation: (**a**) the geometry consists of fluid domains of a perforation and a wellbore and (**b**) half of the geometry is generated because of the symmetry.

**Figure 3 sensors-24-05996-f003:**
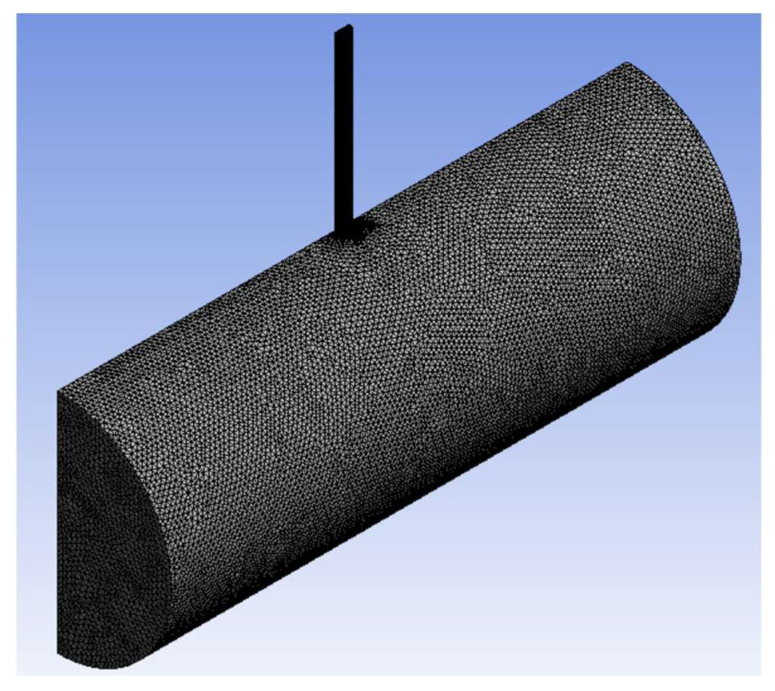
Meshed geometry.

**Figure 4 sensors-24-05996-f004:**
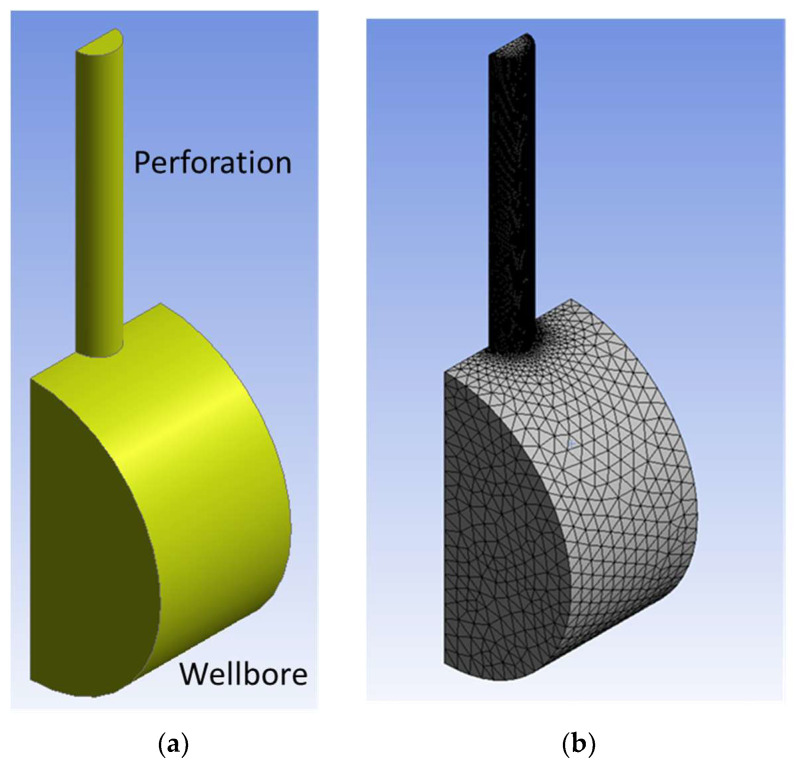
(**a**) Physical geometry and (**b**) the meshed geometry of the small-scale CFD simulation.

**Figure 5 sensors-24-05996-f005:**
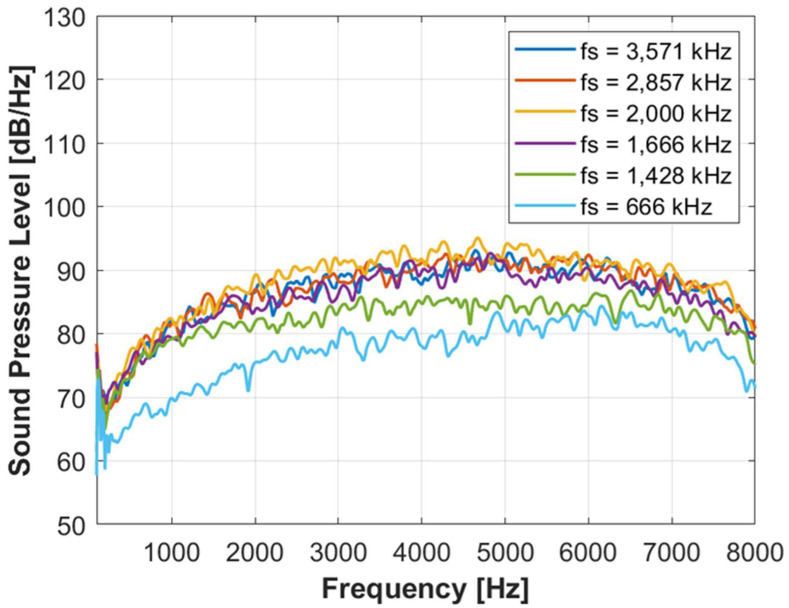
Frequency spectra in the DAS frequency range (less than 8000 Hz) with different sampling frequencies.

**Figure 6 sensors-24-05996-f006:**
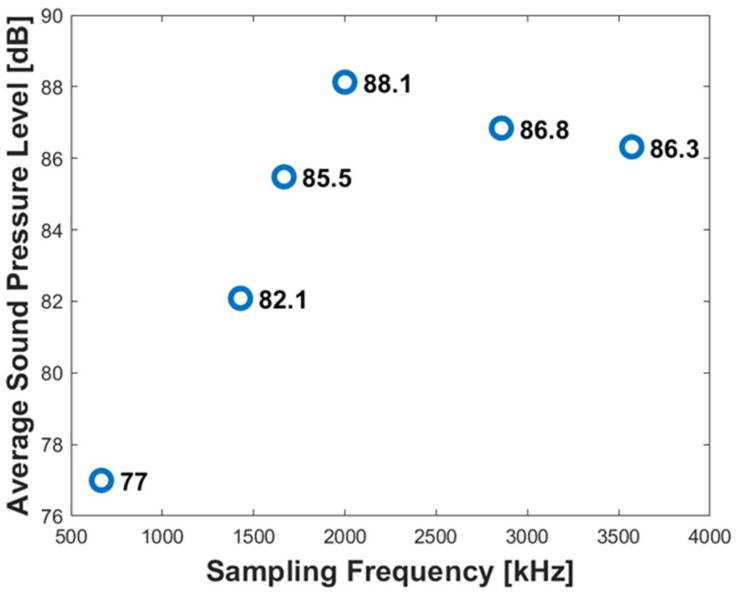
Average sound pressure level depending on sampling frequency.

**Figure 7 sensors-24-05996-f007:**
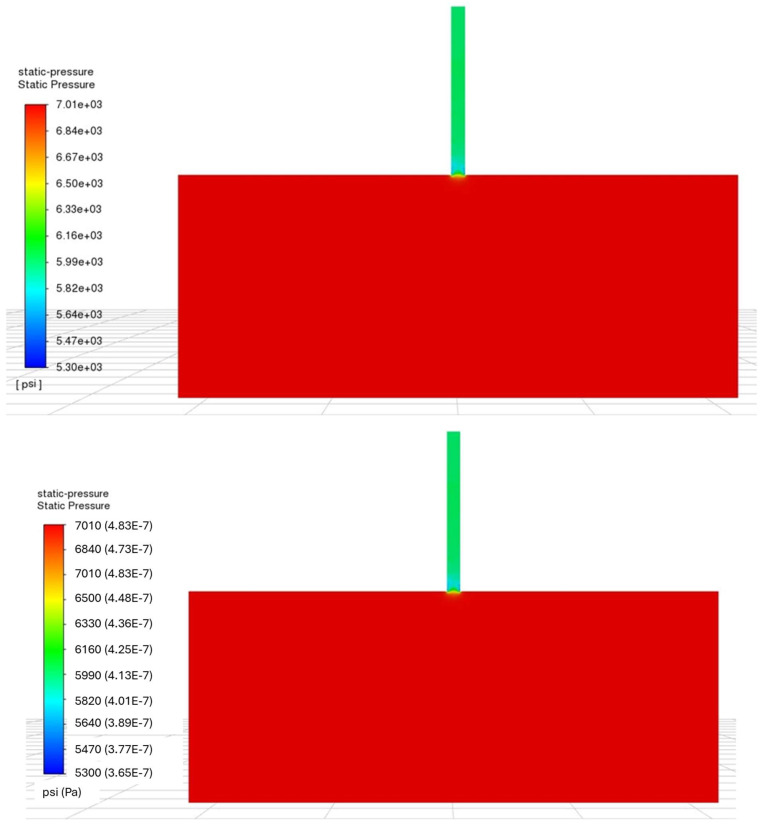
Static pressure calculated by LES.

**Figure 8 sensors-24-05996-f008:**
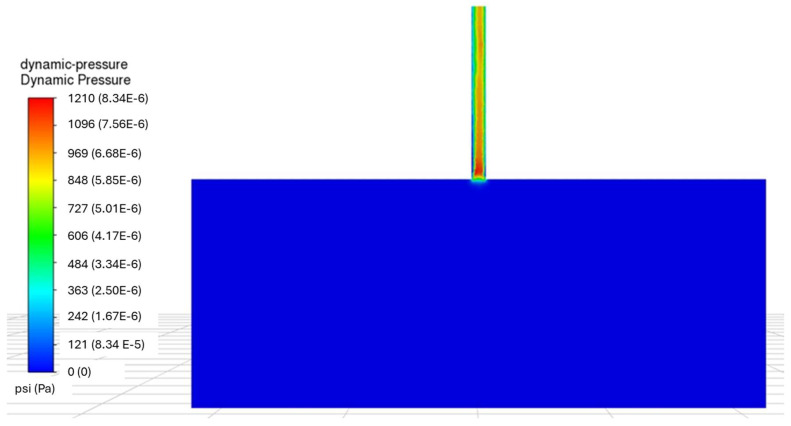
Dynamic pressure calculated by LES.

**Figure 9 sensors-24-05996-f009:**
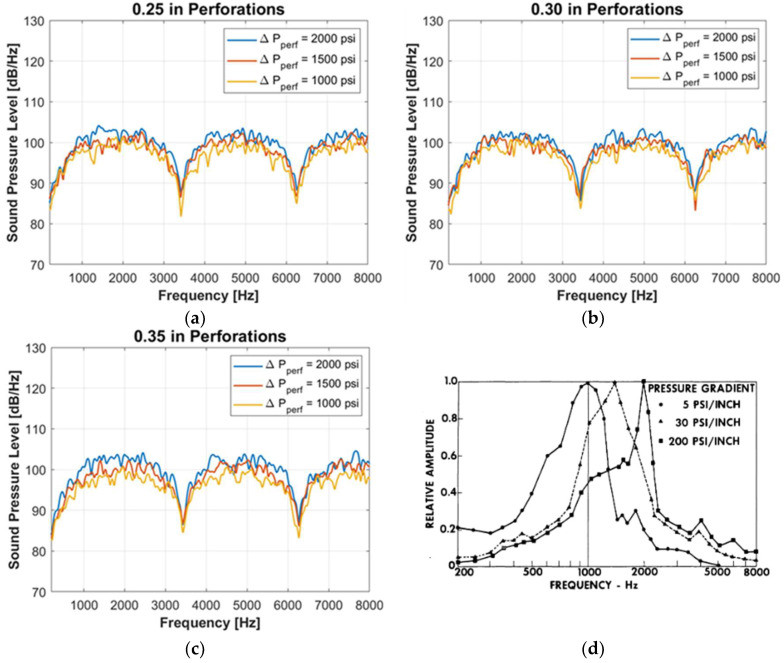
Calculated sound spectra: (**a**) 0.25 inches perforation diameter, (**b**) 0.30 inches perforation diameter, and (**c**) 0.35 inches perforation diameter. Additionally, (**d**) McKinley’s experimental result [13] is shown for comparison.

**Figure 10 sensors-24-05996-f010:**
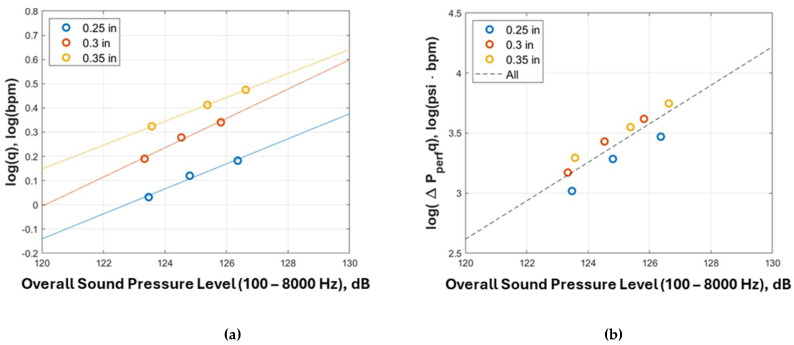
Acoustic correlation for various perforation sizes: overall sound pressure level versus (**a**) $\log\left(q^{3}\right)$, (**b**) $\log\left(∆p_{perf}q\right)$, and (**c**) $\log\left(v^{4}d\right)$.

**Figure 11 sensors-24-05996-f011:**
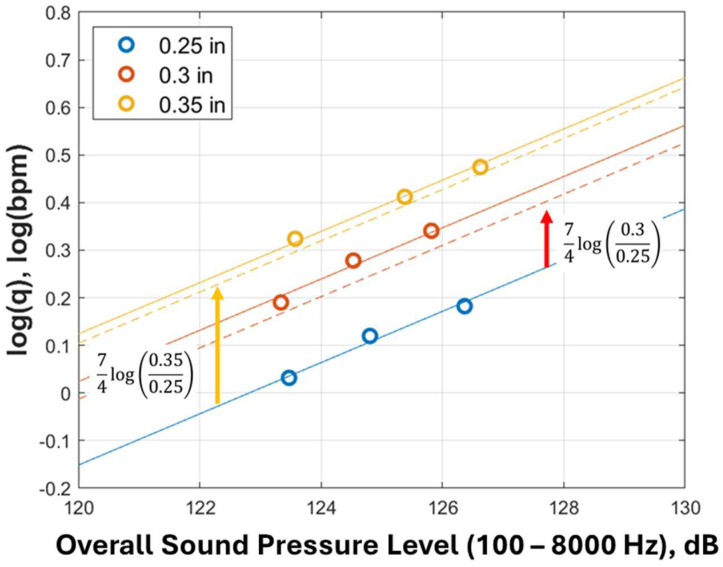
Acoustic correlation compensating perforation erosion effect. Red arrow points to the slope of the red dashed-line, and yellow arrow points the slope of the yellow dashed-line.

**Figure 12 sensors-24-05996-f012:**
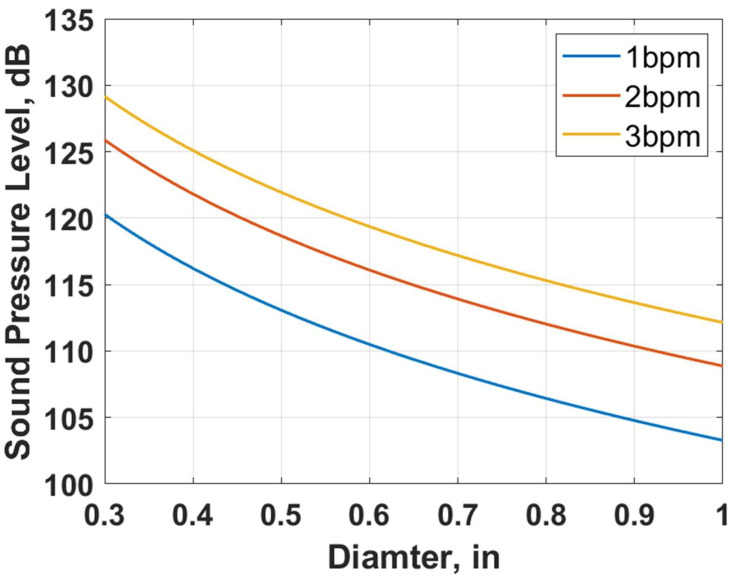
Perforation flow-induced acoustic model sensitivity.

**Table 1 sensors-24-05996-t001:** Six cases evaluated mesh size and sampling frequency.

Parameters	Case 1	Case 2	Case 3	Case 4	Case 5	Case 6
Perforation domain mesh size (in (cm))	0.05 (0.13)	0.03 (0.076)	0.025 (0.064)	0.02 (0.051)	0.01 (0.025)	0.008 (0.020)
Time step size (sec)	1.5 × 10^−6^	7.0 × 10^−7^	6.0 × 10^−7^	5.0 × 10^−7^	3.5 × 10^−7^	2.8 × 10^−7^
Sampling frequency (kHz)	666	1428	1666	2000	2857	3571

**Table 2 sensors-24-05996-t002:** Nine cases of the large-scale simulation with a sharp edge perforation.

Perforation diameter (in (cm))	0.25 (0.64)			0.30 (0.76)			0.35 (0.89)		
Target perforation friction pressure (psi (Pa))	1000 ($6.9\times 10^{6}$)	1500 ($1.0\times 10^{7})$	2000 $\left(1.4\times 10^{7}\right)$	1000 $\left(6.9\times 10^{6}\right)$	1500 ($6.9\times 10^{6}$)	2000 ($1.4\times 10^{7}$)	1000 ($6.9\times 10^{6}$)	1500 ($1.0\times 10^{7}$)	2000 ($1.4\times 10^{7}$)
Injection rate (bbl/min (m^3^/s))	1.08 (0.029)	1.31 (0.035)	1.52 (0.0040)	1.54 (0.0041)	1.89 (0.0048)	2.19 (0.0058)	2.10 (0.0056)	2.58 (0.0068)	2.98 (0.0079)
Sampling frequency (kHz)	2500	2500	2500	2500	2500	2500	2000	2000	2000

## Data Availability

The original contributions presented in the study are included in the article, further inquiries can be directed to the corresponding author.

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
