# Peer review of "Investigation of the Reduction in Distributed Acoustic Sensing Signal Due to Perforation Erosion by Using CFD Acoustic Simulation and Lighthill’s Acoustic Power Law"

_sensors, 2024, doi:10.3390/s24185996_

Round 1

Reviewer 1 Report

Comments and Suggestions for Authors

The authors Hamanaka et al. explored the relationship between perforation erosion and Distributed Acoustic Sensing (DAS) signal attenuation using Computational Fluid Dynamics (CFD) and Lighthill's acoustic power law.However, in the reviewer opinion the paper needs minor revisions to be recommendable for publication.

1. In the introduction section, some description of perforation erosion and its effect on DAS signaling is provided, but the specific statement of the research question could have been made more explicit. It is recommended that the introduction clearly identify the shortcomings of existing methods and clearly describe how this study fills these gaps.

2.  In the introduction, the research background of CFD and vortex topics in recent years should be added. The CFD technology should be referenced. You can reference to literature: 1. Multi-field coupling vibration patterns of the multiphase sink vortex and distortion recognition method. 2.  Key technologies and development trends of the soft abrasive flow finishing method.

3. In the modeling section, the description of mesh-independence is not clear enough; why not analyze mesh size and time step independently of each other?

4. The manuscript refers to the use of the Ffowcs-Williams-Hawkings model for acoustic calculations. It is suggested that the reasons for the choice of this model be detailed, as well as its comparison with other models in terms of accuracy and computational efficiency.

5. The discussion of relevant results in the results section could be further expanded by analyzing the causes of the phenomenon in question and its implications.

6. The concluding section repeats some of the findings and suggests that the main conclusions be summarized and specific directions for future research be proposed.

Comments on the Quality of English Language

No

Author Response

Reviewer 1

  1. In the introduction section, some description of perforation erosion and its effect on DAS signaling is provided, but the specific statement of the research question could have been made more explicit. It is recommended that the introduction clearly identify the shortcomings of existing methods and clearly describe how this study fills these gaps.

Author Response: The introduction is revised to incorporate the comment. The revision includes explain what the application of DAS is, why the current interpretation has weakness and misleading information, leading to the objective of this study and summary of approach.

  1. In the introduction, the research background of CFD and vortex topics in recent years should be added. The CFD technology should be referenced. You can reference to literature: 1. Multi-field coupling vibration patterns of the multiphase sink vortex and distortion recognition method. 2. Key technologies and development trends of the soft abrasive flow finishing method.

Author Response: We have added two references in this section and revised the section as (line 92)

“Vortices caused by multiphase flow are studied across various industries, including aerospace, nuclear power, and manufacturing, using Computational Fluid Dynamics (CFD) modeling. In [11], the acoustics induced by a multiphase sink vortex propagating through multiple acoustic media was modeled by coupling CFD with fluid-solid acoustic theory. Additionally, [12] provides a comprehensive review of the Soft Abrasive Flow (SAF) finishing method, emphasizing modeling efforts that investigate the interaction between abrasive particle flow and the workpiece using CFD.”

  1. In the modeling section, the description of mesh-independence is not clear enough; why not analyze mesh size and time step independently of each other?

Author Response: Thank you for pointing out. It is because the time step size must be limited relative to the mesh size to stabilize the calculation. We have added additional explanation (line 272)

“To stabilize the transient flow calculation, the mesh size is adjusted according to the sampling frequency. This adjustment is necessary because numerical simulations discretize partial differential equations (PDEs) in both time and space. To ensure accuracy, fluid particles must not skip grid points within a single time step, adhering to the Courant–Friedrichs–Lewy (CFL) condition.”

  1. The manuscript refers to the use of the Ffowcs-Williams-Hawkings model for acoustic calculations. It is suggested that the reasons for the choice of this model be detailed, as well as its comparison with other models in terms of accuracy and computational efficiency.

Author Response: Ffowcs-Williams-Hawkings model enables us to consider the sound generated by perforation wall, providing more realistic representation compared to other aeroacoustics models. Additional explanation has been added (line 232)

“Ansys Fluent employs Ffowcs-Williams and Hawkings model, enabling us to calculate sound generated at perforation wall, which Lighthill’s equation cannot account for, as well as sound induced by turbulence.”

  1. The discussion of relevant results in the results section could be further expanded by analyzing the causes of the phenomenon in question and its implications.

Author Response: Additional discussion about frequency shift is added (line 355)

“Figure 9 (a)-(c) do not exhibit clear peaks within the DAS frequency band for any perforation sizes, and the spectra have symmetrical shapes. However, the empirical results published in [13], which examined sound induced at a throttle, demonstrated peak frequencies shifting from 2,000 Hz to 1,000 Hz as the throttle pressure decreases, as shown in Figure 9 (d). Furthermore, [9] observed frequency shifts in field DAS data due to perforation erosion. One reason for the discrepancies with the previous studies could be the small perforation sizes used in the simulation. This is explained by Lighthill’s dimensionless analysis, which relates the characteristic frequency,  where  is the characteristic velocity and  is the characteristic length. In our situation, the characteristic velocity is the velocity at the perforation entrance and the characteristic length is the perforation diameter. Therefore, the minimum characteristic frequency is 10,114 Hz when the perforation pressure is 1,000 psi ( Pa) and the perforation diameter is 0.35 inches (0.89 cm). It is possible that certain peaks were not observed because the characteristic frequencies lie outside the DAS frequency range. However, if the perforation diameter increases or the flow velocity decreases, the characteristic frequency will decrease, and a peak may be observed within the DAS frequency range.

Another potential reason for the discrepancies is that the simulation assumes the acoustic waves propagate through a uniform fluid surrounding the turbulence. In contrast, in the field or experiment, the induced sound propagates through the media with various densities and sound speed, such as pipes, cement and formation, which could cause frequency change.”

  1. The concluding section repeats some of the findings and suggests that the main conclusions be summarized and specific directions for future research be proposed.

Author Response: The finding and conclusion have been revised to make it clear and future research is added to it.

Reviewer 2 Report

Comments and Suggestions for Authors

The paper deals with theoretical CFD calculations carried out on ANSYS on the influence of a perforation on the propagation of acoustic waves. There are some pressure calculations for various flow rates and perforation sizes.

Global remarks on the text:

Some references other than conference papers are missing.

The units should be better defined in the SI (in, bbl….)

There are two meshing figures and no simulation results (pressure, streamlines, etc.), which is a shame...

For minor corrections inside the text:

For Figure 1, which is used as an experimental justification, not all the details are visible, so the information is either displayed legibly or removed.

Can you justify the small range of size of perforation 0,25 to 0,35 (please do not answer computational cost). It could be interesting to track smaller size to have the sensibility of this technique.

Lines 418, 437 and 438, avoid capital letter.

Author Response

Reviewer 2

Global remarks on the text:

  1. Some references other than conference papers are missing.

Author Response: we have added two more references regarding CFD at line 92.

  1. The units should be better defined in the SI (in, bbl….)

Author Response: The SI unit has been provided in brackets following the oil field unit.

  1. There are two meshing figures and no simulation results (pressure, streamlines, etc.), which is a shame...

Author Response: Figures showing static pressure and dynamic pressure have been added.

For minor corrections inside the text:

  1. For Figure 1, which is used as an experimental justification, not all the details are visible, so the information is either displayed legibly or removed.

Author Response: The figure has been revised and now the labels for axis are clear.

  1. Can you justify the small range of size of perforation 0,25 to 0,35 (please do not answer computational cost). It could be interesting to track smaller size to have the sensibility of this technique.

Answer: that is what been used in the field.

  1. Lines 418, 437 and 438, avoid capital letter.

Author Response: They have been corrected to use lowercase letters.